# Production of highly oxygenated organic molecules (HOMs) from trace contaminants during isoprene oxidation

Anne-Kathrin Bernhammer[1,2], Lukas Fischer[1], Bernhard Mentler[1], Martin Heinritzi[3], Mario Simon[3] and Armin Hansel[1,2]

[1]Institute for Ion and Applied Physics, University of Innsbruck, 6020 Innsbruck, Austria
[2]IONICON Analytik GmbH, 6020 Innsbruck, Austria
[3]Institute for Atmospheric and Environmental Sciences, Goethe University of Frankfurt, 60438 Frankfurt am Main, Germany

*Correspondence to*: A. Hansel (armin.hansel@uibk.ac.at)

**Abstract.**

During nucleation studies from pure isoprene oxidation in the CLOUD chamber at CERN we observed unexpected ion signals at $m/z$ = 137.133 ($C_{10}H_{17}^+$) and $m/z$ = 81.070 ($C_6H_9^+$) with the recently developed proton transfer reaction time-of-flight mass spectrometer (PTR3-TOF) instrument. The mass-to-charge ratios of these ion signals typically correspond to protonated monoterpenes and their main fragment. We identified two origins of these signals: first secondary association reactions of protonated isoprene with isoprene within the PTR3-TOF reaction chamber and secondly [4+2] cycloaddition (Diels-Alder) of isoprene inside the gas bottle which presumably forms the favoured monoterpenes limonene and sylvestrene, as known from literature. Under our PTR3-TOF conditions used in 2016 an amount (relative to isoprene) of 2 % is formed within the PTR3-TOF reaction chamber and 1 % is already present in the gas bottle. The presence of unwanted cycloaddition products in the CLOUD chamber impacts the nucleation studies by creating ozonolysis products as corresponding monoterpenes, and is responsible for the majority of the observed highly oxygenated organic molecules (HOMs). In order to study new particle formation (NPF) from pure isoprene oxidation under atmospheric relevant conditions, it is important to improve and assure the quality and purity of the precursor isoprene. This was successfully achieved by cryogenically trapping lower volatility compounds such as monoterpenes before isoprene was introduced into the CLOUD chamber.

## 1 Introduction

Emissions of biogenic volatile organic compounds (BVOCs) impact the oxidation capacity of the atmosphere and serve as precursors for secondary organic aerosols (SOA) (Hallquist et al., 2009) through the formation of low-volatility oxidation products. They are emitted by a large variety of vegetation (~ 1150 Tg C yr$^{-1}$). The most abundantly emitted BVOCs on a global scale are isoprene (70 %) and monoterpenes (11 %) (Guenther et al., 2012). These BVOCs serve as main gas phase precursors for lower volatility oxidation products that play a crucial role for secondary organic aerosol (SOA) formation, and have also been related to new particle formation (NPF) over forest regions in the presence of sulfuric acid (Hallar et al., 2011; Held et al., 2004; Pierce et al., 2014; Pryor et al., 2010; Riipinen et al., 2007; Yu et al., 2014). While their contribution to the

global carbon budget is well established in general, too little is known about the contribution of different BVOC species to NPF.

The role of monoterpenes, especially α-pinene, has been extensively studied in this context. These studies suggest that NPF proceeds via highly oxidised low volatility organic compounds that are formed through autoxidation reactions of peroxy radicals from α-pinene ozonolysis (Ehn et al., 2014; Kirkby et al., 2016; Kulmala et al., 2013; Schobesberger et al., 2013; Riccobono et al., 2014; Winkler et al., 2012). HOMs were shown to nucleate at atmospherically relevant concentrations on their own or with the help of sulfuric acid (Kirkby et al., 2016; Tröstl et al., 2016). Despite being the most dominant BVOC on a global scale (Guenther et al., 2012) the role of isoprene in NPF is far less understood compared to monoterpenes(e.g. Hoffmann et al., 1998; Koch et al., 2000; Bonn et al., 2002; Ehn et al., 2014; Kirkby et al., 2016; Tröstl et al., 2016; Kulmala et al., 2013; Schobesberger et al., 2013; Riccobono et al., 2014; Winkler et al., 2012). The importance of isoprene oxidation products for aerosol formation has been shown in laboratory studies (Claeys et al., 2004; Kroll and Seinfeld, 2008). Products from oxidation with OH radicals (e.g. isoprene epoxydiols (IEPOX)) can actively partition into atmospheric aerosol particles to form SOA (Lin et al., 2013, L. Xu et al., 2015). Epoxides have recently been shown in laboratory studies to contribute to the growth of sulfuric acid particles (Surratt et al., 2010; W. Xu et al., 2014; Budisulistiorini et al., 2017). However, based on the observed suppression of biogenic NPF in the presence of isoprene in plant chamber studies, Kiendler-Scharr et al. (2009) proposed a chemical mechanism that is based on OH depletion by isoprene itself to explain the observed suppression of NPF, and suggested that the suppression effect depends on the concentration ratio ($R$) of isoprene carbon to monoterpene carbon, where an increase in the ratio $R$ leads to a decrease of nucleation rates. While this mechanism may be reasonable in a well-controlled chamber environment with relatively simple chemical processes, especially in terms of $HO_x$ and $NO_x$ chemistry, it is not plausible for real forests. Numerous field studies and atmospheric observations have detected no reduction in OH radical concentration caused by isoprene or any other BVOC (Lelieveld et al., 2008; Kubistin et al., 2010, Martinez et al., 2010, Hansen et al., 2017). Even though OH radical concentrations are not reduced by isoprene, a suppression of NPF is observed in isoprene-dominant forests as a recent summertime field study (SOAS) undertaken in Alabama (southeast of US), Lee et al. (2016) has shown, where the smallest particles do not grow in isoprene-dominated environments despite apparently favourable chemical precursor conditions. A prominent example is the Amazon rainforest ($R \sim 15$) where extensive and continuous aerosol measurements have been conducted over the last decades. These measurements show a consistent lack of NPF at forest sites (Pöhlker et al., 2012) as well as sites influenced by biomass-burning (Rissler et al., 2006). The goal of these studies, however, was not to investigate the chemical mechanism behind the occurring suppression of NPF in the Amazon rainforest. Besides the Amazon, other studies have also reported the absence of NPF in several isoprene dominant forests across the United States (Kanawade et al., 2011; Bae et al., 2010; Pillai et al., 2013; Hallar et al., 2016; Yu et al., 2015). The $R$ values observed during SOAS were between 1 and 10, within the range of other $R$ values from forest with a reported lack of NPF in summertime. In the boreal forests in Hyytiäla, Finland on the other hand, the $R$ value was only ~0.18 and NPF was frequently observed.

The simple algorithms of current climate models predict NPF with nucleation rates independent of the $R$ value since they only take the total sum of low-volatility organic compounds into account, without regard to their actual composition in forests

(Kirkby et al., 2011: Riccobono et al., 2014). As Lee et al. (20016) have demonstrated it is not feasible to simply apply current biogenic NPF knowledge to isoprene-dominant forests because it is derived solely from laboratory experiments of pure monoterpene oxidation. At present, it is still unclear how the oxidation products from isoprene may alter the oxidation chemistry of terpenes and in turn affect NPF in mixed forest environments. Further studies on this subject are required to improve our understanding.

During isoprene NPF studies at the CLOUD chamber, we detected unexpected ion signals at $m/z$ = 137.133 ($C_{10}H_{17}^+$) and $m/z$ = 81.070 ($C_6H_9^+$), besides the expected protonated isoprene, with the recently developed high resolution proton transfer reaction time-of-flight mass spectrometer (PTR3-TOF, Breitenlechner et al., 2017). Here we will explain and discuss the origin of these ion signals that are caused in part by monoterpene like contamination of isoprene. We will give an estimate how big the nucleation rate and the early growth rate are changed due to the presence of monoterpene like contamination. Installing a cryotrap in the isoprene supply line decreased the contamination as well as related oxidation products such as some $C_{10}$ and all $C_{15}$ and $C_{20}$-HOMs significantly.

## 2 Experimental

### 2.1 Instrumentation

#### 2.1.1 PTR3-TOF

In the present study a novel proton transfer reaction-time-of-flight mass spectrometer (PTR-MS) called the PTR3-TOF, that utilises a new gas inlet and an innovative reaction chamber design was used, that is described in detail in (Breitenlechner et al., 2017). The new reaction chamber consists of a tripole operated with RF (radio frequency) voltages generating an electric field only in the radial direction. An elevated electrical field is necessary to reduce clustering of primary hydronium ($H_3O^+$) and product ions with water molecules present in the sample gas. The PTR3-TOF was operated at 80 mbar pressure, and a constant temperature of 38 °C in the tripole reaction chamber. The RF amplitude was adjusted to 700-800 $V_{p-p}$, which corresponds to an E/N of typically 95 Td ($E$, electric field strength; $N$, number gas density; unit, Townsend, $Td$; 1 Td = $10^{-17}$ V cm$^2$). During CLOUD 10 (autumn 2015) and CLOUD 11 (autumn 2016) campaigns the PTR3-TOF was regularly calibrated. For this purpose, known concentrations of isoprene and α-pinene from a gas standard were diluted in 1 slpm zero air. Calibrations were performed for typical operating conditions of the CLOUD chamber: 38 % and 85 % relative humidity at 5 °C. The instrumental background signal was determined by measuring chamber zero air. PTR3-TOF raw data have been processed using newly developed software capable of high resolution and multipeak analysis as described in Breitenlechner et al., 2017. Processed data were duty cycle corrected (Dcps, $(i) = Cps(i) * \sqrt{101/m_i}$ ) which compensates for mass-dependent transmission of the TOF mass spectrometer.

### 2.1.2 CI-APi-ToF

The Chemical Ionisation Atmospheric Pressure Interface Time of Flight mass spectrometer (nitrate-CI-API-TOF, Tofwerk AG, Thun, Switzerland) uses an ion source similar to the design of Eisele and Tanner (1993) although, instead of a radioactive source, a corona discharge is used to generate nitrate primary ions $NO_3^-(HNO_3)_{0-2}$ (Kürten et al., 2011). The instrument was calibrated with respect to sulfuric acid (Kürten et al., 2012), and for the mass dependent transmission efficiency (Heinritzi et al., 2016). For detailed information on the quantification of highly oxygenated organic molecules, the reader is referred to Kirkby et al. (2016).

### 2.1.3 Cryotrap

A cryotrap was added to the isoprene gas supply line directly behind the gas bottle to freeze out possible lower volatility contaminations and effectively remove impurity compounds like monoterpenes or higher oxidised organics. The cryotrap consisted of a chiller coil placed inside a 100 mm diameter dewar flask filled with Huber DW-Therm thermofluid. The chiller maintained the liquid at -57.6 °C. Surrounding the chiller coil and immersed in the dewar liquid was a second spiral coil with six rings of 85 mm diameter through which the gas passed straight from the isoprene bottle (max. flow 10 sccm, average flow rate 5 sccm). The dewar flask was positioned upstream of the isoprene mass flow controller (MFCs). The isoprene coil was 6/4 mm diameter (OD/ID) electropolished stainless steel, and was thoroughly cleaned before use. The total length of the gas pipe inside the dewar flask was approximately 1.8 m. The cryotrap was in use for a total of 14 days measurement time.

## 2.2 Experimental procedures

### 2.2.1 Chamber experiments

Measurements were carried out in the CLOUD (Cosmics Leaving Outdoor Droplets) chamber at CERN (Duplissy et al., 2016; Kirkby et al., 2011) during the CLOUD 10 campaign in autumn 2015 and the CLOUD 11 campaign in autumn 2016. Experiments at the CLOUD chamber are generally carried out in a continuous and dynamical manner instead of separate experiments in "batch" mode.

During nucleation studies of pure isoprene over the course of the CLOUD 10 campaign in 2015, 13.5 ppbv of isoprene (1 % isoprene in $N_2$, purity 99 %, CHARBAGAS AG) were introduced into the chamber. Ozone (45 ppbv) was introduced approximately 10 hours after conditioning the chamber in two steps for additional 9.5 hours. The experiment was carried out at 85 % relative humidity and 5 °C. Additional experiments were conducted during the CLOUD 11 campaign in 2016, this time with a cryotrap added to the isoprene supply line. For this purpose, 33.5 ppbv of isoprene were injected into the chamber, and ozone (35 ppbv) was added 5 hours later. After a reaction time of 5.5 hours the cryotrap was activated to remove low volatility contaminants and measurements continued for another 4 hours. This experiment was carried out at 37 % relative humidity and 6 °C. Gas phase precursors were measured with the PTR3-TOF. Simultaneously, a nitrate chemical-ionisation

atmospheric-pressure-interface time-of-flight (CI-API-TOF) mass spectrometer was used to analyse the highly oxygenated organic compounds from ozonolysis experiments.

### 2.2.2 Cryotrap evaporation experiment

After a total of two weeks of operation, the cryotrap was disconnected from the chamber system and directly connected to the
PTR3-TOF inlet. A steady stream of dry $N_2$ (3 slpm) was fed through the isoprene coil. After a baseline determination, the spiral coil was removed from the chiller liquid and allowed to warm up from -57.6 °C to room temperature over a period of three hours. The concentrations of released organic compounds were measured with the PTR3-TOF. After 45 min. an additional 1:2 dilution with dry $N_2$ became necessary due to a rapid depletion of primary ions.

### 3 Result and discussion

Figure 1a shows the temporal behaviour of the ion signals at $m/z$ = 69.070 ($C_5H_9^+$) and $m/z$ = 41.039 ($C_3H_5^+$), corresponding to protonated isoprene and its fragment, during the 2016 experiment. After isoprene concentrations reached a steady level of ~33.5 ppbv, ozone (~35 ppbv) was continuously injected, and the cryotrap was switched on after several hours of oxidation to freeze out possible low volatility contaminants whose presence was suspected in 2015. As expected, the ion signals at $m/z$ = 69.070 and $m/z$ = 41.039 were linearly correlated over the entire course of the experiment, independent of experimental
conditions, consistent with fragment formation inside the PTR3-TOF instrument (Fig. 1b).

We observed additional ion signals at $m/z$ = 137.133 ($C_{10}H_{17}^+$) and $m/z$ = 81.070 ($C_6H_9^+$) when isoprene was added. These ion signals correspond to typical mass-to-charge ratios of protonated monoterpenes and the corresponding fragment ion. Contrary to the isoprene signal, the ion signals at both $m/z$ = 137.133 and $m/z$ = 81.070 show dependence on ozone, and a decrease induced by the cryotrap. Using the sensitivity of the α-pinene calibration, about 1 ppbv of α-pinene or monoterpene analogues
were observed. This amount is equivalent to 3 % of the measured isoprene (33.5 ppbv). Correlation of $m/z$ = 81.070 and $m/z$ = 137.133 shows two slopes, which, in turn, hints at two different sources for the $C_{10}H_{17}^+$ signal observed with PTR3-TOF (Fig. 1c).

We explain the formation of the "monoterpene" signal, on one hand, with secondary association reactions of protonated isoprene with isoprene within the PTR3-TOF reaction chamber forming $C_{10}H_{17}^+$ ions. As will be shown below, this route
accounts for two-thirds of the total signal detected at $m/z$ = 137.133, which is equivalent to 2% relative to the isoprene signal. On the other hand, one-third of the total $C_{10}H_{17}^+$ signal or 1% (relative to isoprene) is caused by dimerization of isoprene inside the gas bottle to form [4+2] cycloaddition products. The amount of dimerization inside the gas bottle was estimated from a chamber experiment during which the cryotrap was switched off at the beginning, and then turned on for the last hours of measurement. The cycloaddition product is equal to the difference, Δ, between the maximum concentration before ozone
injection and the decrease due to oxidation by ozone and freeze-out caused by the cryotrap (Fig. 1a).

**3.1 Secondary association reactions within the PTR3-TOF reaction chamber**

Figure 2a shows the temporal behaviour of the ion signals at $m/z$ = 137.133 ($C_{10}H_{17}^+$), $m/z$ = 81.070 ($C_6H_9^+$) and $m/z$ = 273.258 ($C_{20}H_{33}^+$) monitored during the warm-up of the cryotrap during an experiment at the end of the 2016 campaign after two weeks of freezing out the low volatility impurities of the isoprene. Immediately after removal of the spiral coil from the cooling liquid,

we observed signals at all three mass-to-charge ratios. At a first glance their temporal behaviour seems to be identical, but a closer look at the correlations reveals significant differences. The protonated monoterpene at $m/z$ = 137.133 and its fragment at $m/z$ = 81.070 show a linear dependency (Fig. 2b), which is expected from ion fragmentation within the PTR3-TOF following pseudo first order kinetics between $H_3O^+ \bullet (H_2O)_n$ (n=0-2) primary ions in reactions with the reactant $C_{10}H_{16}$ (1).

$$H_3O^+(H_2O)_n + C_{10}H_{16} \quad \rightarrow \quad C_{10}H_{17}^+ + (n+1)\ H_2O \tag{1a}$$

$$\rightarrow \quad C_6H_9^+ + C_4H_8 + (n+1)\ H_2O \tag{1b}$$

Comparison of $m/z$ = 137.133 and $m/z$ = 273.258, however, reveals a quadratic dependency (Fig. 2c) and indicates that $C_{20}H_{33}^+$ is the product of the secondary association reaction of $C_{10}H_{17}^+$ with $C_{10}H_{16}$ stabilized in collisions with M in the PTR3-TOF reaction chamber (2).

$$C_{10}H_{17}^+ + C_{10}H_{16} + M \quad \rightarrow \quad C_{10}H_{17}^+(C_{10}H_{16}) + M \tag{2}$$

Contrary to a classical proton-transfer-reaction time-of-flight mass spectrometer (PTR-TOF-MS) (Graus et al. 2010), the PTR3-TOF operates at a much higher drift pressure (80 mbar) and at a longer reaction time (3 ms). This leads to a significant increase in ion molecule collisions inside the reaction zone of the PTR3-TOF which enables secondary association reactions to become visible at reactant concentrations of about 10 ppbv compared to 10 ppmv in classical PTR-MS (Hansel et al. 1995). The results shown in Figure 2 demonstrate that monoterpene like compound are released from the spiral coil during warm-up.

This means that monoterpene like contaminants were present in the isoprene gas bottle.

In Figure 3 we compare the correlation of $m/z$ = 137.133 with $m/z$ = 69.070 (Fig. 3a and 3c) obtained during chamber experiments 2016 without the cryotrap and before ozone injection and the correlation of $m/z$ = 273.258 with $m/z$ = 137.133 (Fig. 3b and 3d) obtained during cryotrap evaporation experiments in 2016. While the correlations of $m/z$ = 273.258 with

$m/z$ = 137.133 in Fig. 3b and 3d show a clear quadratic dependence, the correlation of $m/z$ = 137.133 with $m/z$ = 69.070 in Fig. 3a seems to indicate two overlapping processes: a quadratic one from the secondary association reaction and an additional linear process. A closer look at the lower concentration range where secondary association reaction isn't yet dominant reveals that this is indeed the case (Fig. 3c). Therefore, we identify two processes that create the observed signal at $m/z$ = 137.133:

- *Secondary association reaction within the PTR3-TOF reaction chamber*

$$H_3O^+ \bullet (H_2O)_n + C_5H_8 \rightarrow C_5H_9^+ + (n+1)H_2O \tag{3a}$$
$$C_5H_9^+ + C_5H_8 + M \rightarrow C_{10}H_{17}^+ + M \tag{3b}$$

- *Direct ionisation of a $C_{10}$ precursor*

$$H_3O^+ \bullet (H_2O)_n + C_{10}H_{16} \rightarrow C_{10}H_{17}^+ + (n+1)H_2O \tag{4}$$

**3.2 Dimerization inside the gas bottle**

Dimerization of pure isoprene is known to occur when stored without a stabiliser (0.000017 % per hour at 20 °C, Estevez et al., 2014 and reference therein). It is influenced by pressure and temperature and proceeds via a [4+2] cycloaddition with isoprene acting both as diene and dienophile. This reaction can explain the observed compound at $m/z$ = 137.133 and

$m/z$ = 81.070 ($C_6H_9^+$). Cycloaddition of isoprene has been well documented (Citroni et al., 2007; Compton et al., 1976; Estevez et al., 2014; Groves and Lehrle, 1992; Walling and Peisach, 1958), and thermally-induced polymerisation in the gas phase has recently been shown for heated GC-inlets. The dimerization leads predominantly to six-membered rings, mainly the [4+2] Diels-Alder products sylvestrene and limonene (Estevez et al., 2014 and references therein). This known dimerization is the reason for the addition of $p$-tert-butyl catechol (TBC) as a stabiliser to liquid isoprene (Sigma Aldrich, 99 % purity, 139 ppm

TBC for the used gas standard). Additionally, liquid isoprene may already contain up to 2000 ppm of isoprene dimers upon purchase, as stated in the product specification which is also the case for liquid phase isoprene that is more commonly used as a precursor source in isoprene experiments.

Despite the addition of TBC as stabiliser to prevent polymerisation inside the gas bottle (stainless steel, 2 years old in 2016), 1 % of dimerized isoprene (~350 pptv) could be observed at $m/z$ = 137.133 in 2016. The monoterpene analogues, presumably

the favoured cycloaddition products sylvestrene and limonene (Wang et al., 2013), have much lower vapour pressures than isoprene and are, therefore, effectively removed from the system by the cryotrap (Fig. 1a). We found that the ratio between the total signal at $m/z$ = 137.133 (sum of cycloaddition product from gas bottle and secondary ionic clusters) and isoprene changed between the autumn 2015 campaign (CLOUD 10) and the autumn 2016 campaign (CLOUD 11). It doubled over the course of a year, presumably due to a depletion of the stabiliser and increased dimerization of the isoprene precursor from

1.5 % to a total of 3 %.

**3.3 Impact of impurities**

Due to the presence of double bonds, isoprene and especially monoterpenes show a high reactivity towards ozone. The reaction rate of $d$-limonene with ozone is even faster ($21.1 \times 10^{-17}$ $cm^3$ molecule$^{-1}$ s$^{-1}$) than the corresponding reaction rate of α-pinene ($9.4 \times 10^{-17}$ $cm^3$ molecule$^{-1}$ s$^{-1}$) and both are significantly faster than the one of isoprene ($1.3 \times 10^{-17}$ $cm^3$ molecule$^{-1}$ s$^{-1}$ at 298 K,

Khamaganov and Hites, 2001). However, it is convenient to assume that the reaction rate of sylvestrene is also similar to that of limonene considering the structural similarity of limonene (methyl group in *meta*-position instead of *para*-position). The high reaction rates with ozone makes the contaminants ideal candidates to significantly influence distribution of the resulting oxidation products even at comparatively low concentrations.

The distribution of oxidation products after ozone exposure and, more importantly after cryotrap freeze-out of the low volatility

precursors, was investigated during the 2016 CLOUD measurements.

Figure 4 shows a comparison of selected oxidation products from isoprene ozonolysis and monoterpene ozonolysis as observed during CLOUD experiments. The respective oxidation products from isoprene ozonolysis are easily distinguishable from

oxidation products of the monoterpene analogues. Figure 4a shows oxidation products originating from isoprene as precursor. The compounds show a significant increase upon ozone exposure, but are not affected by the cryotrap due to the high vapour pressure of the precursor. Instead they continue to increase until steady state concentrations are reached. The picture for oxidation products from monoterpene ozonolysis is different. A similar increase after ozone exposure can be observed. However, due to the lower vapour pressure of the precursors, deployment of the cryotrap leads to a decrease in the respective signals of the oxidation products as the precursors are frozen out by the cryotrap and are no longer available for oxidation (Fig. 4b).

Due to the different temporal behaviour discrimination between the respective oxidation products is comparatively simple and shows thatdespite significantly lower concentrations of the low volatility precursor, at least one third of the more than 200 identified signals with $H_3O^+\bullet(H_2O)_n$ as primary reagent ions (up to $m/z = 350$) show the behaviour of monoterpene oxidation products. The total raw signals for monoterpene oxidation products are smaller than for isoprene oxidation products but concentrations are still in the pptv range for a given injection of 33.5 ppbv of isoprene with 1 % of monoterpene contamination relative to isoprene concentrations.

The effect of the cryotrap on the oxidation product distribution was not only observed by means of PTR3-TOF but also by means of CI-APi-TOF mass spectrometry. Figure 5 shows a mass defect plot comparing highly oxygenated molecules (HOMs) before and during the deployment of the cryotrap using CI-APi-TOF data from 2016. Unfortunately, steady state could not be reached for the freeze-out of the monoterpene analogues, so monoterpene oxidation products remain visible. It takes 3 hours to exchange the gases in the CLOUD chamber meaning that after starting to completely remove the monoterpenes from the isoprene supply line 3 hours later 63% of the original steady state monoterpene concentration is still present. Nevertheless, the impact of the partial removal of the monoterpene like impurities is clearly visible.

Removal of the monoterpene contaminants leads to a significant decrease in signal intensity and complete disappearance of the heavier masses. $C_{10}$ compounds appear prominently as the dominating species, with $C_{10}H_{14}O_9$ as the predominant compound, without active cryotrap $C_{15}H_{24}O_x$ and $C_{20}H_{30}O_x$ signals are clearly visible (Fig. 5a). Comparison of the HOM spectra show a complete disappearance of the $C_{15}$ and $C_{20}$ bands upon deployment of the cryotrap and also some $C_{10}$ compounds are significantly reduced, as can be seen in Fig. 5b. $C_2H_5O_3$ becomes the predominant compound after freeze-out. This clearly indicates a significant change in the observed oxidation products, and shows how strongly trace contaminations of reactive $C_{10}$ compounds, even at low concentrations, can impact HOM distribution from isoprene ozonolysis. This is due to the capability of monoterpenes to form HOMs and, to a certain degree, an interference between the oxidation mechanism of isoprene and monoterpenes since both mechanism revolve around $RO_2$ chemistry (Teng et al, 2017; Rissanen et al., 2015) which could affect the resulting closed shell HOM distribution. In a recent paper, Berndt et al. 2018a describe the formation of dimers (HOMs) with (fast) accretion product formation from peroxy radicals: $RO_2 + R'O_2 \rightarrow ROOR' + O_2$. The reactivity of this reaction path increases with increasing functionalization of the $RO_2$ radicals. Highest rate constants were observed for $RO_2$ radicals bearing a hydroxyl and an endo-peroxide group besides the peroxy moiety. In analogy, having isoprene ($C_5$) contaminated

with monoterpene like compounds ($C_{10}$) explains the fast formation of $C_{15}$ compounds from $C_5$-$RO_2$ + $C_{10}$-$RO_2$ accretion reactions. $C_{10}$ closed shell HOMs are produced either by direct oxidation of $C_{10}$ contaminants or by $C_5$-$RO_2$ "self reactions". In another manuscript Berndt et al. 2018b describe in detail the mechanism of product formation from $\alpha$-pinene oxidation under the influence of isoprene. The first CLOUD study involving isoprene oxidation makes intensive use of the cryotrap to clean isoprene from monoterpene like contaminations (Heinritzi et al. 2018).

Earlier studies have already argued that not all HOMs measured by the nitrate CI-API-TOF possess extremely low volatility (Kurten et al., 2016, Tröstl et al., 2016). While a large fraction of $C_{10}$ class molecules may be only Low Volatility Organic Compounds or even Semi Volatile Organic Compounds, basically all $C_{20}$ class molecules fall into the extremely low volatility category, which is suspected to be the most relevant for nucleation and early growth. We thus can assume with some certainty that a missing cryotrap, which leads to unintended $C_{20}$ class HOM formation, directly increases measured nucleation rates and early growth of particles. We estimated the effect on nucleation and growth by quantifying the resulting HOMs with and without an active cryotrap and related them to nucleation rates according to Kirkby et al. 2016 and growth rates according to Tröstl et al. 2016.

The comparison for nucleation rate shows that without a cryotrap we have a total HOM concentration of $1.2 \times 10^7$ $cm^{-3}$ which would result in an approximate nucleation rate J of 1.5 $cm^{-3}$ $s^{-1}$. With an active cryotrap, total HOM concentration is reduced to $2.6 \times 10^6$ $cm^{-3}$ which results in a nucleation rate J of $6.5 \times 10^{-2}$ $cm^{-3}$ $s^{-1}$. Performing nucleation experiments without a cryotrap would lead to an overestimation of J by a factor of 23! Thus, isoprene would wrongly be considered as a molecule that is capable of producing pure biogenic nucleation at atmospherically relevant concentrations, while in reality it is not.

Comparison for growth rates ($GR$) using the parameterization from Tröstl et al. 2016 for particles in the 1.7 – 3 nm range and an assumed particle size of 3 nm shows a growth rate of 1.5 nm $h^{-1}$ without active cryotrap and a growth rate of 0.2 nm $h^{-1}$ with active cryotrap. Hence performing growth experiments without a cryotrap would lead to an overestimation of growth rates by approximately an order of magnitude. Thus, isoprene would be attributed to possess a much larger influence on early particle growth while in truth a significant fraction of growth is caused by oxidation products from the contaminants.

## 4 Conclusion

We have observed ion signals at $m/z$ = 137.133 and $m/z$ = 81.070 during presumably pure isoprene oxidation experiments which correspond to monoterpene signals. The sources of these signals were attributed to secondary association reaction between protonated isoprene and isoprene in the PTR3-TOF (two-thirds of total signal) and, more significantly, dimerization of isoprene inside the gas bottle (one third of total signal). While the first result is important for the growing group of PTR3-TOF user, the latter result is important for a much greater group of atmospheric scientists. Isoprene dimer contamination of 2000 ppm is stated by Sigma-Aldrich for their liquid isoprene with purity > 99%. Dimerization of isoprene leads to compounds

identical to monoterpenes in structure and chemical behaviour. The presence of reactive monoterpene like compounds significantly impact the oxidation product distribution. The overall effect of the contaminants has been clearly shown in the mass defect plot. The disappearance of higher masses, especially in the $C_{15}$ to $C_{20}$ range upon deployment of a cryotrap has a profound impact on nucleation and growth rates. An overestimation of – in our case – at least one order of magnitude may thus lead to a misinterpretation of resulting data and its atmospheric implications due to attribution of properties to isoprene that the compound in reality does not possess and are caused by the lower volatile contaminants.

Results of this study, fortunately, do not affect previous CLOUD results as none of them pertain to isoprene effects on nucleation or growth. However, future isoprene studies at the CLOUD chamber will take these findings fully into account and only use data that was obtained with a cryotrap installed in the isoprene supply line.

The findings of the present work can be extended to experiments using evaporation of liquid isoprene as a precursor source. Evaporation of liquid isoprene requires active heating (Dommen et al., 2009), which, unless very carefully controlled, may lead to an increased polymerisation within the source liquid as well as an enhanced evaporation of contaminants into the experimental chamber. As has been shown in this study, it is therefore of vital importance to assure purity of the isoprene precursor when assessing HOM formation, nucleation and growth. Otherwise the influence of lower volatility contaminants on experimental results cannot be ruled out and may lead to misinterpretation of actual atmospheric implications. The required proper precursor control can successfully be achieved via e.g. deployment of a cryotrap upstream of the experimental chamber, as demonstrated in this study.

**Acknowledgments:**

We would like to thank CERN for supporting CLOUD with important technical and financial resources, and for providing a particle beam from the CERN Proton Synchrotron. We also thank Albin Wasem/CERN and Serge Mathot/CERN for designing and constructing the cryotrap. This research has received funding from the EC Seventh Framework Programme (Marie Curie Initial Training Network "CLOUD-TRAIN" no. 316662) and the Austrian Research Funding Association (FFG, Project Number 846050).

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

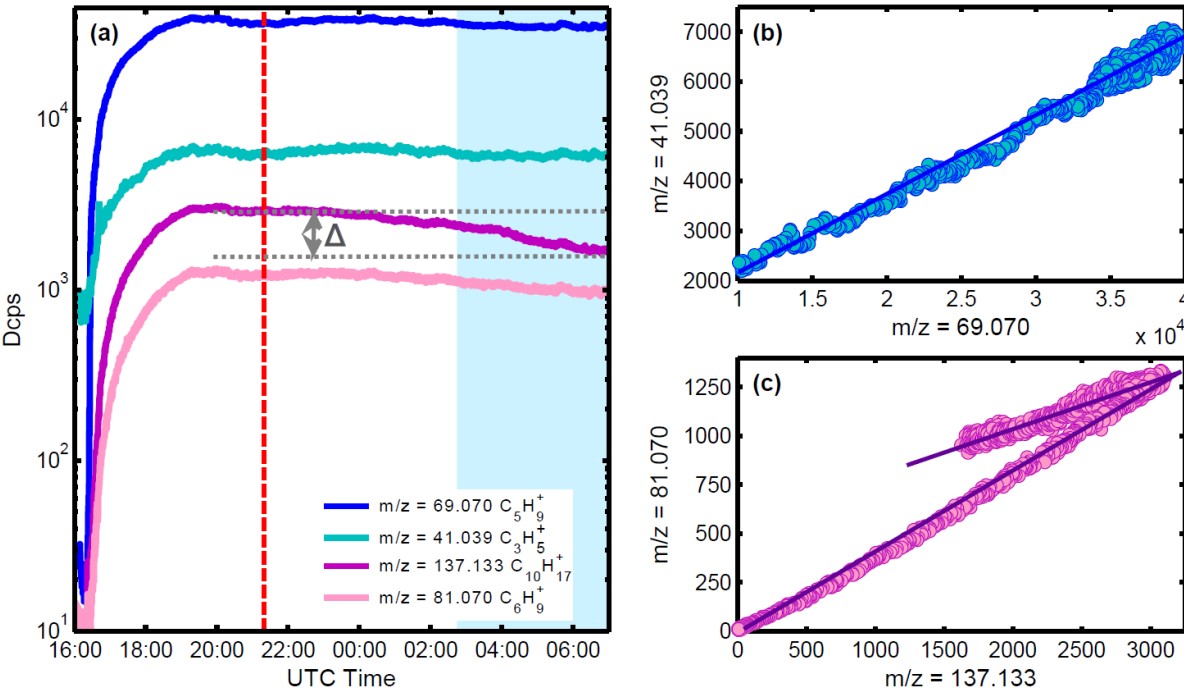

**Figure 1: a)** Time series of isoprene observed at the protonated mass $m/z = 69.070$, its main fragment at $m/z = 41.039$, isoprene cluster/monoterpene ($m/z = 137.133$) and their main fragment ($m/z = 81.070$) during the 2016 experiment. The blue shaded area corresponds to times with the cryotrap switched on, the dashed red line indicates the start of $O_3$, $\Delta$ marks the signal loss caused by ozonolysis and cryotrap freeze-out. **b)** Correlation plot of $m/z = 69.070$ and its main fragment $m/z = 41.039$, **c)** correlation plot of $m/z = 137.133$ and its main fragment $m/z = 81.070$.

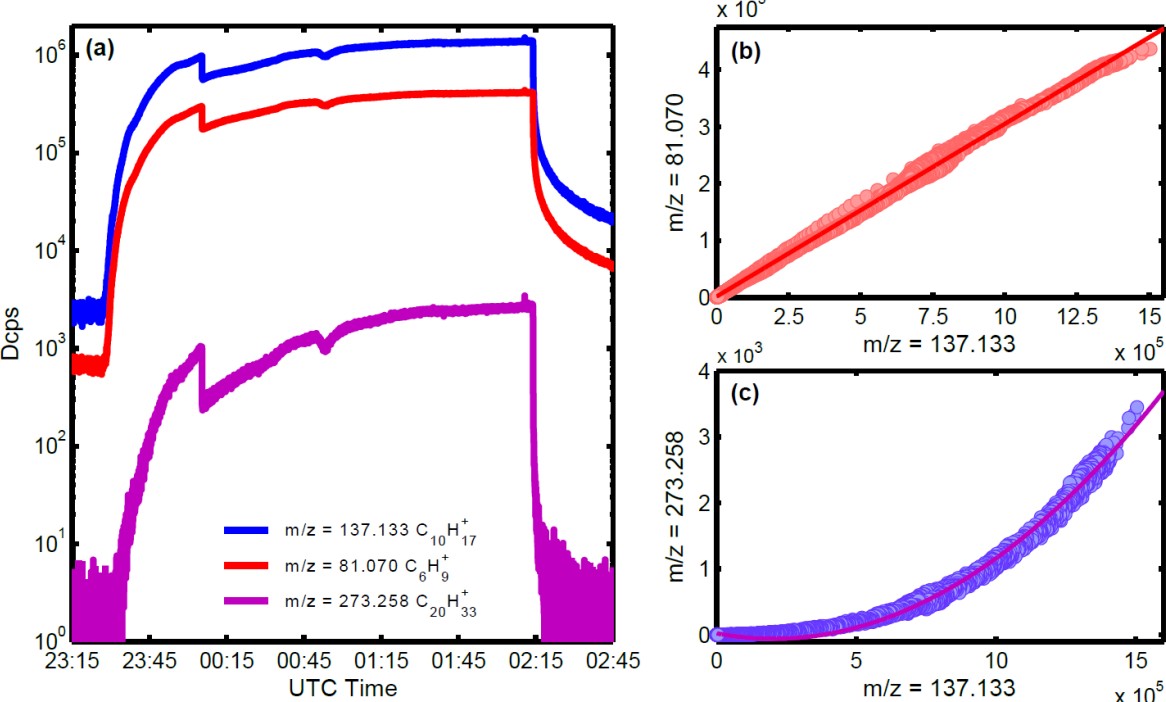

**Figure 2: a)** Time series of $m/z = 137.133$ ($C_{10}H_{17}^+$), $m/z = 81.070$ ($C_6H_9^+$) and $m/z = 273.258$ ($C_{20}H_{33}^+$) during cryotrap evaporation experiment (2016), **b)** correlation plot of $m/z = 137.133$ ($C_{10}H_{17}^+$) and its main fragment $m/z = 81.070$ ($C_6H_9^+$), **c)** correlation plot of $m/z = 137.133$ ($C_{10}H_{17}^+$) and the secondary ionic cluster $m/z = 273.258$ ($C_{20}H_{33}^+$).

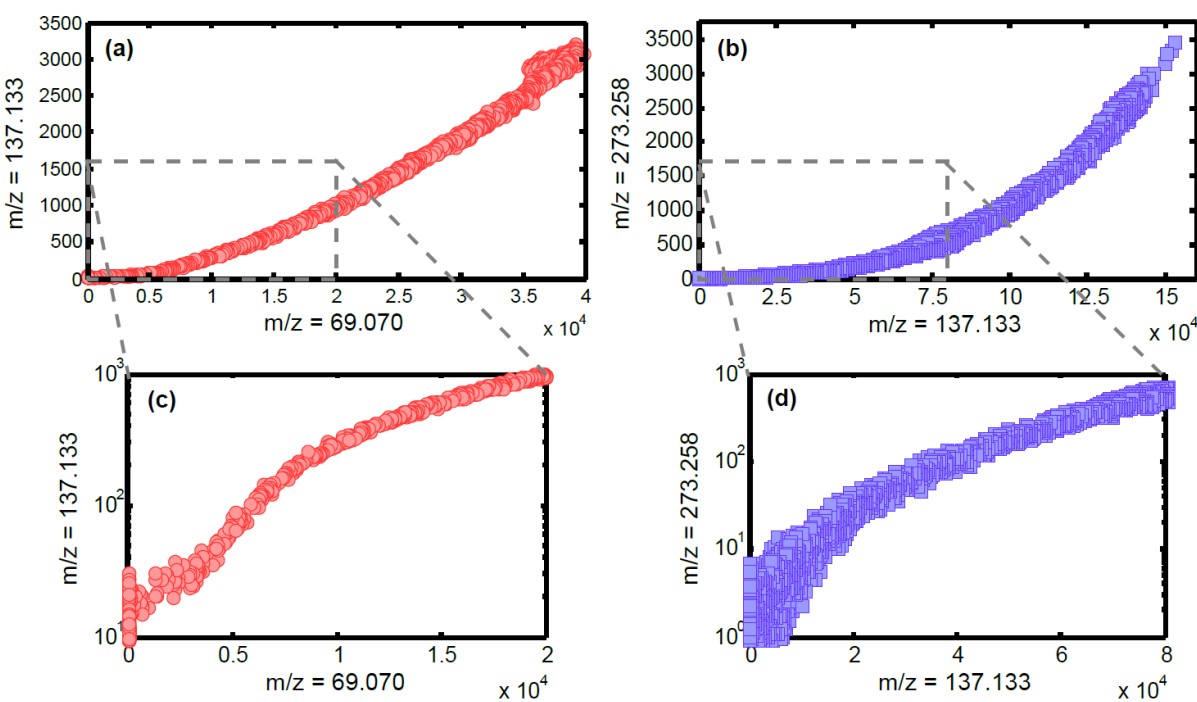

**Figure 3: Comparison of the correlation plots of the 2016 chamber experiment (a) and 2016 evaporation experiment (b) for m/z = 69.070 vs. m/z = 137.133 (red) and m/z = 137.133 vs. m/z = 273.258 (purple). Panels a) shows the correlation over the initial chamber experiment without cryotrap and before the addition of ozone, panel b) shows the correlation over the entire evaporation experiment, panels c) and d) focus on the lower concentration range. For the pure secondary association reaction (b,d) a clear quadratic dependency is observed, while for the chamber experiment (a,c) two overlapping processes – especially in the lower concentration regime – take place before the cluster formation becomes the dominant pathway at higher concentrations.**

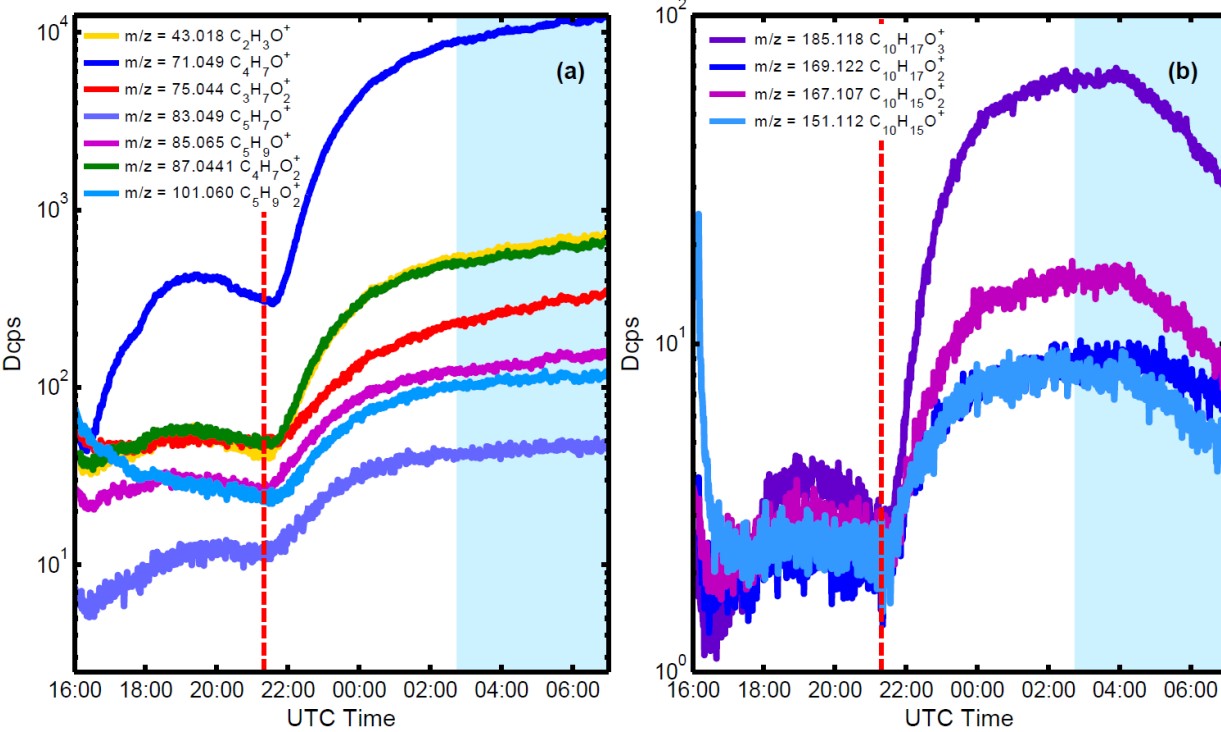

**Figure 4: a) Time series of selected pure isoprene oxidation products in 2016, b) time series of main monoterpene-analogue oxidation products which show a typical decrease towards the end due to a removal of the precursor with the cryotrap while the isoprene products are not affected. The red line corresponds to the injection time of ozone and the blue shaded area to the times with cryotrap.**

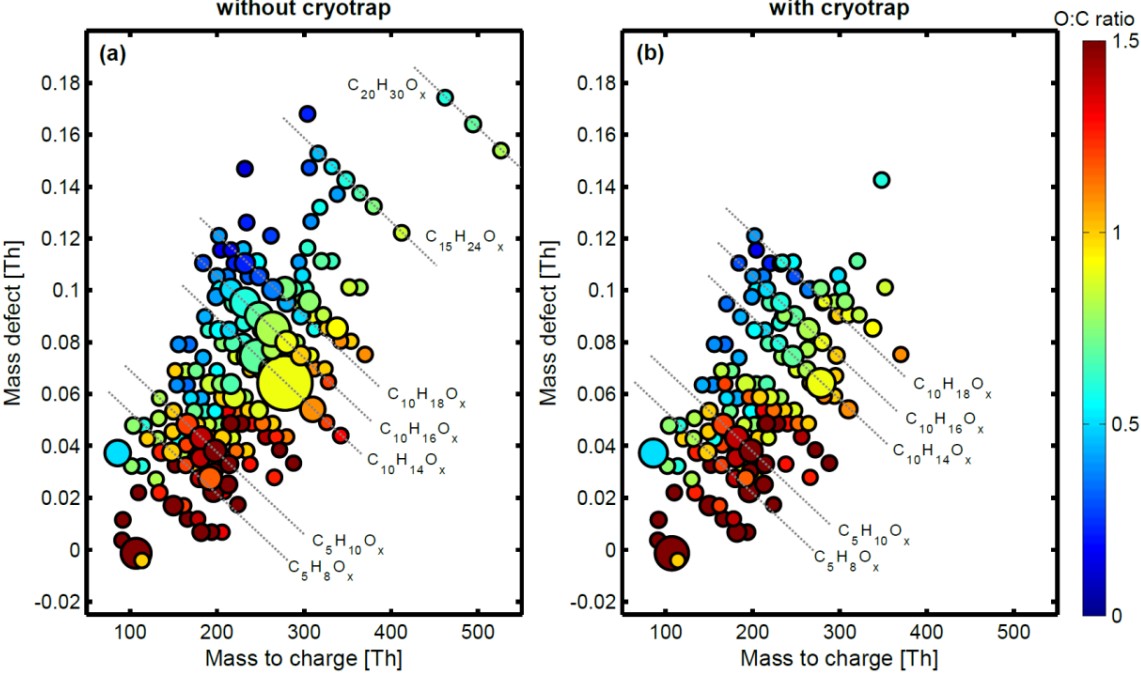

**Figure 5: Mass defect plot obtained from 2016 CI-APi-ToF data after subtraction of primary ions (NO$_3^-$(HNO$_3$)$_{0-2}$) for a) isoprene ozonolysis without cryotrap and b) isoprene ozonolysis with cryotrap. The circle size corresponds to the signal intensity. Steady state was not reached for the measurements with cryotrap therefore cryogenic removal was incomplete and oxidation products from the monoterpene oxidation are still visible. Nevertheless, a significant decrease in signal intensities, especially in the C$_{10}$ range, can be observed.**