# Peer review of "Production of highly oxygenated organic molecules (HOMs) from trace contaminants during isoprene oxidation"

_Atmospheric Measurement Techniques, 2017_

## Referee Comment (RC1) · Anonymous Referee #1 · 30 Jan 2018

This paper reports on ion signals at m/z 137.133 ($C_{10}H_{17}^+$) and m/z 81.070 ($C_6H_9^+$) measured by the proton transfer reaction time-of flight mass spectrometer (PTR3-TOF, Ionicon) during isoprene oxidation experiments in the CLOUD chamber at CERN. These ions correspond to protonated monoterpenes and their fragments and are therefore unexpected / unwanted for pure isoprene oxidation experiments. Authors explain the origin of these compounds by the reactions inside the PTR3 reaction chamber and by cycloaddition of isoprene in the gas bottle itself. Placing cryogenic trap between the gas bottle and the CLOUD chamber shows clear decrease of the signal for these ions and therefore their successful removal. It points out and identifies the source of monoterpene contaminants and the way how to remove them which is important and

valuable information for future experiments involving isoprene. The manuscript itself needs minor revisions prior to being published, at the moment it is clumsy and needs to be more straight-forward. The main point of the paper should revolve around the possible misinterpretation of data when conducting experiments with isoprene, which is why this journal is appropriate for this paper. The biggest issue with the paper comes from stating that the presence of contaminants impacts the gas phase reactions of isoprene. The final sentence of the paper says "This clearly indicates a significant change in the observed oxidation products, and shows how strongly trace contaminations, even at low concentrations, can impact gas phase oxidation processes and the formation of HOMs from isoprene ozonolysis." The first part is true, contaminants can affect the observed oxidation products, but there is no evidence for the later point that contaminants can "impact gas phase oxidation processes and the formation of HOMs from isoprene." How will contaminants stop gas phase processes occurring with isoprene? This could only occur if all of the ozone present is being consumed by reactions with monoterpenes prior to reacting with isoprene. In these experiments this is likely not true. You could argue that dimers formed between an isoprene oxidized product and an alpha-pinene oxidized product impact the gas phase distribution of isoprene oxidation products, but that is not done here. The first point is the important aspect of the paper, contaminants change the observed oxidation products which is remarkably important for possible misinterpretation of the data. The paper as a whole should be geared more towards these efforts, including the introduction. Further, in the introduction it is mentioned the lack of knowledge about the role of isoprene in new particle formation as well as the importance of the ratio of isoprene to alpha-pinene, however it is not discussed in the paper to what extent the monoterpene contaminants would influence the general results from the CLOUD experiments. In Figure 5. you show quite dramatic change in the gas phase composition with/without cryotrap. I assume the nucleation/growth rates must also differ. If so, could you please comment on this even if it lies slightly behind the scope of Atmospheric Measurement Techniques, it might highlight the relevance of your findings.
Minor Comments:

1. Page 1 line 19. Please define New Particle Formation before simply mentioning NPF

2. Page 2 line 3. references are needed since extensive studies are referenced

3. Page 2 line 3-5. references are needed since studies are referenced.

4. Page 2 line 5 "can occur in the presence of sulfuric acid, as well as in its absence" this is a bit weird formulation

5. Page 2 line 8-9. reference studies on NPF of monoterpenes.

6. Page 2 line 13. Epoxide growth on acidic particles

7. Page 2 line 16. explain the concentration ratio (R) and its impact on the NPF because its value is used later on page 2, but no figure of merit is offered. For instance, how does the suppression depend on R? A brief explanation here will help the reader understand the implication of R discussed below.

8. Page 2, line 19. "numerous field studies" yet you mention only one

9. Page 3 line 7. add a sentence that distills the message of the paper.

10. Page 3 line 11. reword to "... a novel proton transfer reaction-time-of-flight mass spectrometer (PTR-MS), called the PTR3-TOF, that utilizes a..." Also, stick to a normal naming convention for the PTR3 it is either called the PTR3 or PTR3-TOF throughout the manuscript.

11. Page 3 line 13. the abbreviation RF is not defined and should be capitalized.

12. Page 3 line 16. see above

13. Page 7 line 6. why not mention how old the bottle was in 2016?

14. Figure 4a and 4b: changes these plots so the legend is not obscuring the traces.

15. Figure 4a: why does the C4H7O+ signal increase prior to O3 addition?

16. Figure 5a: what is the series of points that sits above the C10H18Ox? Is that a C15?

17. Page 7 line 14-28. This section is clumsily put together. It would be clearer to introduce what you want to show prior to showing the figures. This section is all about the effect of the cyrotrap on the oxidation products after the precursor(s) are exposed to ozone.

Set the stage for this at the beginning of the section, and then talk about each figure. The discussion about the rates of reaction of different precursors probably isn't necessary without more discussion about its importance. What point is trying to be made about the rates of reactions with ozone?

18. Page 8 line 6-7: what is the predominant compound after freeze-out?

---

## Referee Comment (RC2) · Anonymous Referee #2 · 31 Jan 2018

Summary

This short work examines isoprene precursor purity during recent CLOUD campaigns at CERN. The authors detect monoterpenes during what were intended to be pure isoprene experiments. They propose that reactions in the PTR3 instrument source account for 2/3 of the detected monoterpene, and the remaining 1/3 to Diels Alder cycloaddition of the gas-phase isoprene cylinder.

The results in this manuscript are technically solid, and it is well-written. But it does not seem to be a completely fleshed out manuscript, and I have reservations about how relevant the research is to the greater atmospheric science community

[Figure]

Major comments

The authors make an excellent summary point: that scientists conducting laboratory experiments should control the purity of their precursor. But I find that the example used in this manuscript is a specific situation of limited importance. Bernhammer et al., claim here that 2/3 of the monoterpene formed from isoprene is due to the unique high pressure (∼80 mbar) of the PTR3 drift cell. But the vast majority of PTR instruments maintain a drift cell ∼ 2 mbar (de Gouw and Warneke), which would make this in-source reaction a consideration only to the 3 PTR3 instruments in existence. Furthermore, the authors (and references therein) suggest that the remaining 1/3 of the observed monoterpenes form directly from the isoprene isoprene precursor in the gas-phase and not the condensed phase. But many isoprene laboratory experiments, particularly in "batch" mode, are conducted by evaporating liquid-phase isoprene (Paulot et al.) into a chamber.

While the CLOUD experiments are influential and important, the authors leave the effects of this work on previous CLOUD results totally unexplored. It is interesting to see that HOMs formed from the contaminants, but how has that affected other CLOUD nucleation studies? Are there other isoprene works that have suffered from this contamination? Why is this specific issue one of interest to the general community. This work would be much stronger if the authors could explore the atmospheric implications of their results.

Minor comments/typos

Figure 1: The pink and purple traces are very difficult to distinguish. Could you please change the color of one of them? P1L13: Should be "these signals: first secondary" P1L29: "have also been" P3L21 "respectively" is unnecessary here P4L4 "to freeze out possible lower volatility contamination" P7L14 comma after "bonds"

References:

de Gouw, J. & Warneke, C. Measurements of volatile organic compounds in the earth's atmosphere using proton-transfer-reaction mass spectrometry. Mass Spectrom. Rev. 26, 223–257 (2007). Paulot, F. et al. Unexpected Epoxide Formation in the Gas-Phase Photooxidation of Isoprene. Science (80-. ). 325, 730–733 (2009).

---

## Referee Comment (RC3) · Anonymous Referee #3 · 2 Feb 2018

This paper describes issues arising from trace contaminants from monoterpenes during isoprene oxidation experiments at the CLOUD chamber. It is clear that to understand isoprene chemistry and its aerosol yield, no contaminants that have a much larger yield than isoprene itself can be present in the chamber, so preparations for the experiments have to be done with great care, particularly on such large scale and important experiments as the CLOUD measurements.

The paper describes that a monoterpene signal was detected using a PTR3 instrument; and it turned out that 2/3 of the signal were due to cluster formation in the PTR3 reaction chamber and 1/3 was an actual impurity in the CLOUD chamber due to

limonene and sylvestrene formation in the isoprene standard. This impurity could be removed using a cryotrap in the inlet for the chamber.

The first part of the impurity signal caused by the high pressure and long reaction times of the PTR3 are more of a curiosity of this specific instrument and could be easily avoided by using a different PTR-TOF instrument or by changing the conditions in the PTR3 to run closer to standard PTR-MS instruments.

The second part of the signal comes from a real impurity, monoterpenes produced in the isoprene standard. As expected, this impurity results in a large number of oxidation products during the ozonolysis and after removing most of the impurity using a cryotrap the additional oxidation products are significantly reduced.

This is the conclusion of this paper, but unfortunately the implications for past results or the interpretation of the isoprene oxidation processes are not discussed. This discussion would be the actual main interest to the scientific community. While the issues discussed here are very important for the measurements during the CLOUD experiments and they need to be discussed and resolved, they are not relevant to the wider scientific community. I simply do not think that this manuscript includes enough scientifically relevant information to warrant publication in AMT and I recommend rejecting the current manuscript without including a solid discussion about the implications on previous and future research on isoprene oxidation.

The manuscript is generally pretty well written. The only issue I want to mention is that it is not clear to well into the manuscript that the experiments seem to be run dynamically and not in a batch mode. This should be mentioned in the description of the CLOUD experiment early on in the manuscript. I had a few other minor comments, but those were all covered by the other reviewers.

---

## Author Comment (AC1) · 4 May 2018

We would like to thank the reviewers for their comments and suggestions that helped to improve the manuscript. In the response below, the reviewer's questions are reproduced in *blue* and our replies are shown in black.

**Anonymous Referee #1**

This paper reports on ion signals at m/z 137.133 (C10H17+) and m/z 81.070 (C6H9+) measured by the proton transfer reaction time-of flight mass spectrometer (PTR3-TOF, Ionicon) during isoprene oxidation experiments in the CLOUD chamber at CERN. These ions correspond to protonated monoterpenes and their fragments and are therefore unexpected / unwanted for pure isoprene oxidation experiments. Authors explain the origin of these compounds by the reactions inside the PTR3 reaction chamber and by cycloaddition of isoprene in the gas bottle itself. Placing cryogenic trap between the gas bottle and the CLOUD chamber shows clear decrease of the signal for these ions and therefore their successful removal. It points out and identifies the source of monoterpene contaminants and the way how to remove them which is important and valuable information for future experiments involving isoprene. The manuscript itself needs minor revisions prior to being published, at the moment it is clumsy and needs to be more straight-forward.

The main point of the paper should revolve around the possible misinterpretation of data when conducting experiments with isoprene, which is why this journal is appropriate for this paper. The biggest issue with the paper comes from stating that the presence of contaminants impacts the gas phase reactions of isoprene. The final sentence of the paper says "This clearly indicates a significant change in the observed oxidation products, and shows how strongly trace contaminations, even at low concentrations, can impact gas phase oxidation processes and the formation of HOMs from isoprene ozonolysis." The first part is true, contaminants can affect the observed oxidation products, but there is no evidence for the later point that contaminants can "impact gas phase oxidation processes and the formation of HOMs from isoprene." How will contaminants stop gas phase processes occurring with isoprene? This could only occur if all of the ozone present is being consumed by reactions with monoterpenes prior to reacting with isoprene. In these experiments this is likely not true. You could argue that dimers formed between an isoprene oxidized product and an alpha-pinene oxidized product impact the gas phase distribution of isoprene oxidation products, but that is not done here.

The first point is the important aspect of the paper, contaminants change the observed oxidation products which is remarkably important for possible misinterpretation of the data. The paper as a whole should be geared more towards these efforts, including the introduction.

Further, in the introduction it is mentioned the lack of knowledge about the role of isoprene in new particle formation as well as the importance of the ratio of isoprene to alpha-pinene, however it is not discussed in the paper to what extent the monoterpene contaminants would influence the general results from the CLOUD experiments.

In Figure 5. you show quite dramatic change in the gas phase composition with/without cryotrap. I assume the nucleation/growth rates must also differ. If so, could you please comment on this even if it lies slightly behind the scope of Atmospheric Measurement Techniques, it might highlight the relevance of your findings.

**Referee **#1** states that the "biggest issue with the paper comes from stating that the presence of contaminants impacts the gas phase reactions of isoprene" and asks "How will contaminants stop gas phase processes occurring with isoprene?".**

The monoterpene like contaminations can of course undergo direct oxidation steps and, as has been shown by various studies, are able to form HOMs. However, there can be a certain degree of interference of isoprene and monoterpene contamination oxidation processes. This is due to the fact that both mechanisms evolve around RO2 radical chemistry (Teng et al., 2017, Rissanen et al., 2015) and termination reactions might occur involving both radicals originating from isoprene as well as from monoterpene contamination oxidation. This clearly affects the resulting closed shell HOM distribution. In a recent paper, Berndt et al. 2018a describe the formation of dimers (HOMs) with (fast) accretion product formation from peroxy radicals: RO2 + R'O2 -> ROOR' + O2. The reactivity of this reaction path

increases with increasing functionalization of the RO2 radicals. Highest rate constants were observed for RO2 radicals bearing a hydroxyl and an endo-peroxide group besides the peroxy moiety. In analogy, having isoprene (C5) contaminated with monoterpene like compounds (C10) explains the fast formation of C15 compounds from C5-RO2 + C10-RO2 accretion reactions. C10 closed shell HOMs are produced either by direct oxidation of C10 contaminants or by C5-RO2 "self reactions".

We have just submitted a manuscript about accretion product formation of  $\alpha$ -pinene and the influence of isoprene where the mechanism is discussed in detail. (Berndt et al. 2018b).

The first CLOUD study involving isoprene oxidation (Heinritzi et al. 2018) uses the cryotrap to clean isoprene from contamination. In said manuscript, we discuss in detail how the presence of isoprene supresses monoterpene induced nucleation.

**The referee points out that our paper should evolve more around the change in observed oxidation products and the possible misinterpretation resulting from that effect.**

We have changed our manuscript accordingly. Additionally, we tried to estimate the impact of the cryotrap on nucleation and early growth rates. Unfortunately, our experiments were performed at the very end of a CLOUD campaign and no particle counting instruments were available at that time. However, we used measured HOM concentrations with and without the cryotrap and calculated nucleation rates according to Kirkby et al. 2016 and growth rates according to Tröstl et al. 2016. Without a cryotrap we measure a total HOM concentration of  $1.2 \times 10^7$  cm-3, which results in an approximate nucleation rate J of 1.5 cm-3s-1. With the cryotrap switched on the total HOM concentration is reduced to  $2.6 \times 10^6$  cm-3, which corresponds to a nucleation rate J of  $6.5 \times 10^{-2}$  cm-3s-1. Performing an isoprene nucleation experiment without a cryotrap would lead to an overestimation of J by a factor of 23! This means that the contaminants, rather than isoprene, contribute to the nucleation rate.

Using the parameterization from Tröstl et al. 2016 we calculate a growth rate of 1.5 nm  $h^{-1}$  without the cryotrap in contrast to a growth rate of 0.2 nm  $h^{-1}$  with the cryotrap for 3 nm particles. Hence performing early growth experiments without a cryotrap would also lead to an overestimation of growth rates by a factor 7 to 8. Thus, the impact of isoprene on nucleation and early growth would lead to a strong overprediction, if isoprene is contaminated as was the case.

The referee also states that "it is not discussed in the paper to what extent the monoterpene contaminants would influence the general results from the CLOUD experiments." We have added a statement to the manuscript that not previously published CLOUD results are affected by our findings, as none of them contain any isoprene effects on nucleation or growth.

**Minor Comments:**

1. Page 1 line 19. Please define New Particle Formation before simply mentioning NPF - done

2. Page 2 line 3. references are needed since extensive studies are referenced – References are added.

3. Page 2 line 3-5. references are needed since studies are referenced – References are added.

4. Page 2 line 5 "can occur in the presence of sulfuric acid, as well as in its absence" this is a bit weird formulation – The sentence now reads "HOMs were shown to nucleate at atmospherically relevant concentrations on their own or with the help of sulfuric acid."

5. Page 2 line 8-9. reference studies on NPF of monoterpenes. – References were added.

6. Page 2 line 13. Epoxide growth on acidic particles – Additional references were added.

7. Page 2 line 16. explain the concentration ratio (R) and its impact on the NPF because its value is used later on page 2, but no figure of merit is offered. For instance, how does the suppression depend on R? A brief explanation here will help the reader understand the implication of R discussed below. – The sentence now reads "…suggested that the suppression effect depends on the concentration ratio (R) of isoprene carbon to monoterpene carbon, where an increase in the ratio R leads to a decrease of nucleation rates."

8. Page 2, line 19. "numerous field studies" yet you mention only one – References were added.

*9. Page 3 line 7. add a sentence that distills the message of the paper.* – The sentence now reads: "Here we will explain and discuss the origin and the impact of these ion signals, highlighting especially the profound impact of potential contaminants on increased HOM concentrations."

10. Page 3 line 11. reword to "... a novel proton transfer reaction-time-of-flight mass spectrometer (PTR-MS), called the PTR3-TOF, that utilizes a ..." Also, stick to a normal naming convention for the PTR3 it is either called the PTR3 or PTR3-TOF throughout the manuscript. – done, consistently changed to PTR3-TOF.

11. Page 3 line 13. the abbreviation RF is not defined and should be capitalized. - done

12. Page 3 line 16. see above - done

*13. Page 7 line 6. why not mention how old the bottle was in 2016?* – The year should have read 2016 instead of 2017. The typo was corrected

14. Figure 4a and 4b: changes these plots so the legend is not obscuring the traces. - done

15. Figure 4a: why does the C4H7O+ signal increase prior to O3 addition? – We have looked into the issue but could not find a conclusive explanation for the signal increase. It coincides with changes in temperature and RH within the CLOUD chamber. However, these changes in experimental conditions are only in the range of a few percent and are unlikely to be the source of the signal increase.

16. Figure 5a: what is the series of points that sits above the C10H18Ox? Is that a C15? – One series of points is  $C_{15}H_{24}O_x$  and the higher one is  $C_{20}H_{30}O_x$ . Figure 5 and the corresponding section was updated accordingly.

17. Page 7 line 14-28. This section is clumsily put together. It would be clearer to introduce what you want to show prior to showing the figures. This section is all about the effect of the cyrotrap on the oxidation products after the precursor(s) are exposed to ozone. Set the stage for this at the beginning of the section, and then talk about each figure. The discussion about the rates of reaction of different precursors probably isn't necessary without more discussion about its importance. What point is trying to be made about the rates of reactions with ozone? – The section has been rewritten. The discussion about the reaction rate is included to highlight that, despite a comparatively low concentration, monoterpene contaminations can still have a significant impact on oxidation product distribution. *18. Page 8 line 6-7: what is the predominant compound after freeze-out*? - C2H3O5 is the predominant compound after freeze-out?

**Anonymous Referee #2**

This short work examines isoprene precursor purity during recent CLOUD campaigns at CERN. The authors detect monoterpenes during what were intended to be pure isoprene experiments. They propose that reactions in the PTR3 instrument source account for 2/3 of the detected monoterpene, and the remaining 1/3 to Diels Alder cycloaddition of the gas-phase isoprene cylinder. The results in this manuscript are technically solid, and it is well-written. But it does not seem to be a completely fleshed out manuscript, and I have reservations about how relevant the research is to the greater atmospheric science community.

**Major comments**

The authors make an excellent summary point: that scientists conducting laboratory experiments should control the purity of their precursor. But I find that the example used in this manuscript is a specific situation of limited importance. Bernhammer et al., claim here that 2/3 of the monoterpene formed from isoprene is due to the unique high pressure (~80 mbar) of the PTR3 drift cell. But the vast majority of PTR instruments maintain a drift cell~2 mbar (de Gouw and Warneke), which would make this in-source reaction a consideration only to the 3 PTR3 instruments in existence.

Furthermore, the authors (and references therein) suggest that the remaining 1/3 of the observed monoterpenes form directly from the isoprene isoprene precursor in the gas-phase and not the condensed phase. But many isoprene laboratory experiments, particularly in "batch" mode, are conducted by evaporating liquid-phase isoprene (Paulot et al.) into a chamber.

While the CLOUD experiments are influential and important, the authors leave the effects of this work on previous CLOUD results totally unexplored. It is interesting to see that HOMs formed from the contaminants, but how has that affected other CLOUD nucleation studies? Are there other isoprene works that have suffered from this contamination?

Why is this specific issue one of interest to the general community. This work would be much stronger if the authors could explore the atmospheric implications of their results.

We agree with referee #2 that in-source reactions of the scale reported in our manuscript are unique to the PTR3-TOF, which uses 80 mbar in the reaction chamber. However, the PTR3-TOF is a very new and promising instrument, so we regard a careful characterisation important for PTR3-TOF users and also for other CIMS instruments using higher pressure attempting to measure precursor compounds. The dimerization from the diels alder reaction could have been observed by a classical PTR-MS. On the other hand, PTR3-TOF has been designed to measure first and higher order oxidation products as well. Here we could demonstrate that contaminants impact nucleation and early growth more than the precursor isoprene.

The referee further states that many other experiments use liquid isoprene. However, according to the data sheet for liquid isoprene with purity >99 % that is provided by Sigma-Aldrich

(https://www.sigmaaldrich.com/catalog/product/aldrich/464953?lang=de&region=DE) we find information on the addition of *p-tert*-butylcatechol as inhibitor (100 - 150 ppm), as already stated in the manuscript, as well as an upper limit of isoprene dimer contamination of 2000 ppm. This is in the same contamination level that we find for our gas bottle. The use of liquid isoprene without further purification is no guarantee that unwanted contaminants such as isoprene dimers are absent. Our paper provides detailed gas phase measurements as well as a performance test of a cryogenic trap to resolve this issue. Thus, we consider it not only of interest for the CLOUD community.

So far there are no published CLOUD studies that investigate the effect of isoprene oxidation products on nucleation and growth, so all previously published CLOUD results remain unaffected by our findings. But our findings reported here are of vital importance for future publications. E.g. Heinritzi, M., et al., in preparation, 2018, Berndt et al., submitted, 2018b

As described in the response to referee #1 we tried to estimate the impact of contaminants on nucleation and early growth and found that both, nucleation rate and early growth rate, are overestimated by a factor of 23 and 7-8, respectively. We have included these impacts in our manuscript to underline the broader atmospheric relevance.

Minor comments/typos

Figure 1: The pink and purple traces are very difficult to distinguish. Could you please change the color of one of them? – done, changed to a lighter pink P1L13: Should be "these signals: first secondary" - done P1L29: "have also been" - done P3L21 "respectively" is unnecessary here - done P4L4 "to freeze out possible lower volatility contamination" - done P7L14 comma after "bonds" - done References: de Gouw, J. & Warneke, C. Measurements of volatile organic compounds in the earth's atmosphere using proton-transfer-reaction mass spectrometry. Mass Spectrom. Rev. 26, 223–257 (2007). Paulot, F. et al. Unexpected Epoxide Formation in the Gas-Phase Photooxidation of Isoprene. Science (80-. ). 325, 730–733 (2009).

**Anonymous Referee #3**

This paper describes issues arising from trace contaminants from monoterpenes during isoprene oxidation experiments at the CLOUD chamber. It is clear that to understand isoprene chemistry and its aerosol yield, no contaminants that have a much larger yield than isoprene itself can be present in the chamber, so preparations for the experiments have to be done with great care, particularly on such large scale and important experiments as the CLOUD measurements. The paper describes that a monoterpene signal was detected using a PTR3 instrument; and it turned out that 2/3 of the signal were due to cluster formation in the PTR3 reaction chamber and 1/3 was an actual impurity in the CLOUD chamber due to limonene and sylvestrene formation in the isoprene standard. This impurity could be removed using a cryotrap in the inlet for the chamber.

The first part of the impurity signal caused by the high pressure and long reaction times of the PTR3 are more of a curiosity of this specific instrument and could be easily avoided by using a different PTR-TOF instrument or by changing the conditions in the PTR3 to run closer to standard PTR-MS instruments. The second part of the signal comes from a real impurity, monoterpenes produced in the isoprene standard. As expected, this impurity results in a large number of oxidation products during the ozonolysis and after removing most of the impurity using a cryotrap the additional oxidation products are significantly reduced. This is the conclusion of this paper, but unfortunately the implications for past results or the interpretation of the isoprene oxidation processes are not discussed. This discussion would be the actual main interest to the scientific community. While the issues discussed here are very important for the measurements during the CLOUD experiments and they need to be discussed and resolved, they are not relevant to the wider scientific community. I simply do not think that this manuscript includes enough scientifically relevant information to warrant publication in AMT and I recommend rejecting the current manuscript without including a solid discussion about the implications on previous and future research on isoprene oxidation.

The manuscript is generally pretty well written. The only issue I want to mention is that it is not clear to well into the manuscript that the experiments seem to be run dynamically and not in a batch mode. This should be mentioned in the description of the CLOUD experiment early on in the manuscript. I had a few other minor comments, but those were all covered by the other reviewers.

We thank referee #3 for commenting on our manuscript. The main objection the referee raises is that we do not discuss "the implications for past results or the interpretation of the isoprene oxidation processes ". There are no past results from the CLOUD experiment that are affected by isoprene contamination issues. So far, CLOUD has only published nucleation studies that consider monoterpene oxidation products or inorganic precursors. There is a manuscript in preparation (Heinritzi et al. 2018) that makes full use of the cryotrap, as the current manuscript has pointed out the importance of isoprene contamination with respect to HOM composition.

Secondly, we did not make any interpretation of isoprene oxidation processes within the CLOUD experiment that were misguided by a missing cryotrap. Instead, the only interpretation that was made is that without a cryotrap there is an absolutely non-negligible contamination issue that has to be resolved prior to drawing any further scientific conclusions from measured isoprene oxidation data. The proof of effective removal of contaminations is provided in this manuscript. The mentioned upcoming manuscript on isoprene effects on nucleation takes this into account, as it only uses periods with fully active cryotrap. As stated in our reply to referee #1, we discussed the interference of isoprene and monoterpene oxidation processes and estimated the subsequent consequences for nucleation and growth in this manuscript, but this is definitely beyond the scope of the current manuscript.

The referee writes that our paper is lacking a "solid discussion about the implications on previous and future research on isoprene oxidation". As stated, there is no previous research on isoprene oxidation that is affected by a missing cryotrap in CLOUD and all future research is using and will use a cryotrap. This manuscript however describes the important steps necessary to ensure a clean isoprene injection into a chamber and clearly shows the impact on the highly oxygenated molecules present in the CLOUD chamber. As pointed out in the answer to referee #2, the issue of isoprene dimer contamination is not limited to isoprene stored in gas bottles, but also concerns chamber experiments where isoprene is evaporated into the chamber from the liquid phase. Taking this into account we would strongly argue that our findings are of relevance for a wider scientific community, i.e. every experiment that conducts isoprene oxidation, nucleation or SOA studies.

**References:**

Berndt, T., Scholz, W., Mentler, B., Fischer, L., Herrmann, H., Kulmala, M., and Hansel, A., Accretion product formation from self- and cross-reactions of RO2 radicals in the atmosphere, Angew. Chemie. Int. Ed. 10.1002/anie.201710989, 2018a

Berndt et al., submitted, 2018b

Kirkby, J., Duplissy, J., Sengupta, K., Frege, C., Gordon, H., Williamson, C., Heinritzi, M., Simon, M., Yan, C., Almeida, J., Tröstl, J., Nieminen, T., Ortega, I. K., Wagner, R., Adamov, A., Amorim, A., Bernhammer, A.-K., Bianchi, F., Breitenlechner, M., Brilke, S., Chen, X., Craven, J., Dias, A., Ehrhart, S., Flagan, R. C., Franchin, A., Fuchs, C., Guida, R., Hakala, J., Hoyle, C. R., Jokinen, T., Junninen, H., Kangasluoma, J., Kim, J., Krapf, M., Kürten, A., Laaksonen, A., Lehtipalo, K., Makhmutov, V., Mathot, S., Molteni, U., Onnela, A., Peräkylä, O., Piel, F., Petäjä, T., Praplan, A. P., Pringle, K., Rap, A., Richards, N. A. D., Riipinen, I., Rissanen, M. P., Rondo, L., Sarnela, N., Schobesberger, S., Scott, C. E., Seinfeld, J. H., Sipilä, M., Steiner, G., Stozhkov, Y., Stratmann, F., Tomé, A., Virtanen, A., Vogel, A. L., Wagner, A. C., Wagner, P. E., Weingartner, E., Wimmer, D., Winkler, P. M., Ye, P., Zhang, X., Hansel, A., Dommen, J., Donahue, N. M., Worsnop, D. R., Baltensperger, U., Kulmala, M., Carslaw, K. S., and Curtius, J.: Ion-induced nucleation of pure biogenic particles, Nature, 533, 521–526, doi:10.1038/nature17953, 2016.

Heinritzi, M., et al., in preparation, 2018

- Kürten, A., Bergen, A., Heinritzi, M., Leiminger, M., Lorenz, V., Piel, F., Simon, M., Sitals, R., Wagner, A. C., and Curtius, J.: Observation of new particle formation and measurement of sulfuric acid, ammonia, amines and highly oxidized organic molecules at a rural site in central Germany, Atmos. Chem. Phys., 16, 12793-12813, https://doi.org/10.5194/acp-16-12793-2016, 2016.
- Rissanen, M. P., Kurtén, T., Sipilä, M., Thornton, J. A., Kausiala, O., Garmash, O., Kjaergaard, H. G., Petäjä, T., Worsnop, D. R., Ehn, M., and Kulmala, M.: Effects of chemical complexity on the autoxidation mechanisms of endocyclic alkene ozonolysis products: From methylcyclohexenes toward understanding α-pinene, The journal of physical chemistry. A, 119, 4633–4650, doi:10.1021/jp510966g, 2015.
- Teng, A. P., Crounse, J. D., and Wennberg, P. O.: Isoprene Peroxy Radical Dynamics, Journal of the American Chemical Society, 139, 5367–5377, doi:10.1021/jacs.6b12838, 2017.
- Tröstl, J., Chuang, W. K., Gordon, H., Heinritzi, M., Yan, C., Molteni, U., Ahlm, L., Frege, C., Bianchi, F., Wagner, R., Simon, M., Lehtipalo, K., Williamson, C., Craven, J. S., Duplissy, J., Adamov, A., Almeida, J., Bernhammer, A.-K.,

Breitenlechner, M., Brilke, S., Dias, A., Ehrhart, S., Flagan, R. C., Franchin, A., Fuchs, C., Guida, R., Gysel, M., Hansel, A., Hoyle, C. R., Jokinen, T., Junninen, H., Kangasluoma, J., Keskinen, H., Kim, J., Krapf, M., Kürten, A., Laaksonen, A., Lawler, M., Leiminger, M., Mathot, S., Möhler, O., Nieminen, T., Onnela, A., Petäjä, T., Piel, F. M., Miettinen, P., Rissanen, M. P., Rondo, L., Sarnela, N., Schobesberger, S., Sengupta, K., Sipilä, M., Smith, J. N., Steiner, G., Tomè, A., Virtanen, A., Wagner, A. C., Weingartner, E., Wimmer, D., Winkler, P. M., Ye, P., Carslaw, K. S., Curtius, J., Dommen, J., Kirkby, J., Kulmala, M., Riipinen, I., Worsnop, D. R., Donahue, N. M., and Baltensperger, U.: The role of low-volatility organic compounds in initial particle growth in the atmosphere, Nature, 533, 527–531, doi:10.1038/nature18271, 2016.

---

## Author Response (AR2)

Dear Editor,
We would like to thank all referees.
As only referee #5 raised some comments and suggestions (in black), please find enclosed our response (in blue). We hope that the manuscript can now be published in AMT.

With kind regards, Armin Hansel

Anonymous Referee #2

accepted as is

Anonymous Referee #4

accepted as is

Anonymous Referee #5

This review reports on observations of 10 to 30-carbon containing ions formed detected in the new PTR3-TOF instrument from Ionicon during isoprene + O3 experiments conducted at the CERN CLOUD chamber in 2015 and 2016. They conclude that part (2/3) of C10 containing species detected before oxidation arise from secondary ion/molecule reactions of protonated isoprene with additional isoprene within the PTR3 instrument due to the high pressure and long reaction times. The remainder (1/3) of signal is proposed to arise from C10 compounds present in the gas-phase isoprene standard (gas cylinder) formed from diels alder cycloaddition type reactions of isoprene. They show that the C10 compounds can be at least partially removed from the isoprene by passing the gas through a cold trap (-57C), and more importantly, that these impurities constituted a significant fraction of the highly oxygenated material (HOM) formed during the O3 + isoprene experiments.

While the manuscript is not substantially different from the initial submission, I will argue that this type of paper has an important place in the scientific literature. All too often 'negative' type results are not reported due to idea that these are not publishable. While the results published here could easily be placed within another paper alongside more scientifically interesting ideas, there is no reason in today's electronic age that they cannot be also published as standalone technical notes. Not reporting such results leads to others repeating and rediscovering (or worse, not discovering!) the same problems. So, after consideration of the following points, I suggest this manuscript is appropriate for publication in AMT.

Main comments:

Introduction: Own it: I suggest the authors truly commit to writing this type of paper. [Sub-text: Re HOM yield measurements: one needs to be extremely careful regarding purity of precursor, as small impurity high HOM yielding species can have major impact on inferred yield of target species]. Instead of the introduction focusing on how isoprene may suppress HOM formation in the atmosphere – something the rest of the paper really has little to nothing to do with – limit introduction to discussion of what HOM are, why they are important, how they are formed, and why/how very tiny (<1%) impurities in precursor can significantly impact results. What about impurities in other compounds, like say terpenes?

The introduction was rewritten according to the reviewer suggestions to put the focus on HOMs.

Add a figure showing the diels alder dimerization with identified products… with your detection methods and you conclusively identify products? Or do you only get molecular formula?

PTR-TOF technology only allows the measurement of molecular formulas.

Offline GC/MS analysis confirmed the presence of nine monoterpene isomers inside the gas bottle which could not be specified due to lack in standards.

We have discussed these results and cite the relevant literature concerning diel alder dimerization.

Quantification: Manuscript discusses C10 impurities relative to isoprene as a fraction (molar?), while figures show normalized ion counts. The calibration procedure, how one moves from one to the other, needs to be discussed.

This was done, the calibration procedure was described and sensitivities are now included in the 2.1.1 PTR-TOF part.

Add figure, or perhaps a panel to Figure 4 of HOM timeline for experiment shown in figure 4. Show total HOM, and major HOM components.

Figure 4c (panel) was added. (see below)

[Figure]

Specific comments:

P2 L6: define HOM. – done

P2 L20-21: Suggest removal of this sentence. This issue is seemingly unresolved and mounting evidence to the contrary.

The Introduction was rewritten as suggested.

P3 L1: typo '20016' – done

P6 LN20: Maybe… but this statement is somewhat speculative, as is. I.E. Some C10 species could have been introduced into the bottle when standard was made. Or some dimer could be made in regulator on way to instrument? Perhaps expand this discussion to include more possibilities and your lines of reasoning for excluding certain pathways.

The isoprene standard was made by an external supplier (see below) and came with a certificate of analysis, stating the isoprene purity.  Section 3.2

We have included GC/MS results from the isoprene bottle showing that 9 monoterpene isomers were present in our bottle.

The total monoterpene impurities increased within one year in the same bottle from 2015 to 2016. We speculate that this is caused by dimerization reactions within the gas bottle. Dimerization was observed by other groups through diel alder reactions, which is referenced.

P7 LN9-10: This statement as written is somewhat confusing: Are you saying that you made the gas standard from liquid isoprene containing 139 ppm TBC? If so, why not include a complete description of the standard; when it was made, how it was made. How it was certified, etc. Note, that some people when working with isoprene use only the vapor over the liquid to use in experiments (presumably this contains much less impurities, than complete evaporations). What type of bottle was this standard stored in? How was standard certified as a function of time? Did you measure the stabilizer with the PTR? – All useful details to include.

This was done.

In detail:

The gas standard was prepared by an external supplier, Carbagas, using purest nitrogen and liquid isoprene (≥99 % purity) from Sigma Aldrich. According to the Isoprene certificate of analysis it contained 139 ppm of TBC and had a purity of 99.8 % according to GC analysis. It is the same kind of liquid isoprene (https://www.sigmaaldrich.com/catalog/product/aldrich/464953?lang=de®ion=DE) that other studies use as gas (head space from that liquid). The standard was not certified as a function over time as it's use was still well within the guaranteed timeframe for stability as confirmed by the supplier. However an offline GC analysis of the gas bottle was carried out after CLOUD 10, confirming the presence of TBC as well as the presence of nine different monoterpene isomers. A more conclusive identification of the monoterpene isomers was not done due to lack of standards.

The sentence was changed to: "*This known dimerization is the reason for the addition of p-tert-butyl catechol (TBC) as a stabiliser to the liquid isoprene (Sigma Aldrich, ≥99 % purity, with addition of 139 ppm TBC) which was used for creation of the gas standard provided by the supplier. The stabiliser itself could not be identified by PTR3-TOF due to an interference with major monoterpene oxidation products (C10H14O-H+).*"

P8 L9: insert space between 'that' and 'despite' – done

P8 L29: add 's' to mechanism – done

P8 LN27-30: C10 produces HOM in described experiments appears well established here; however, the second part, is not established well. Does the C10 species impact the HOM production from real isoprene in these experiments significantly? Without further data/quantification included, I'm not sure this can be concluded.

We added the following sentence: *The impact of the C10 contaminant is most prominently visible in the significant decrease of C10 compounds and the disappearance of C15 and C20 compounds upon activation of the cryotrap as shown in fig. 5b where the mentioned formation pathway via said accretion reactions is no longer feasible and less HOMs are formed.*

P9 LN4-5: Is there a better place for this sentence? Seems out of place. Remove 'intensive'. – done

P9 LN14-18: Something amiss with numbers here: higher HOM concentration yields lower J? Also, J1/J2 doesn't give 23? – done.

P9 LN30: add 's' to 'user' – done

P10 L1: word 'identical' is used here but throughout the paper 'monoterpene-like' is used. These seem inconsistent with each other. Either they C10 species are monoterpenes or they aren't

It reads now "monoterpenes" throughout the paper.